# Extracellular Electron Transfer in Microbiologically Influenced Corrosion of 201 Stainless Steel by *Shewanella algae*

**DOI:** 10.3390/ma16155209

**Published:** 2023-07-25

**Authors:** Weiwei Chang, Xiaohan Wang, Huaibei Zheng, Tianyu Cui, Hongchang Qian, Yuntian Lou, Jianguo Gao, Shuyuan Zhang, Dawei Guo

**Affiliations:** 1National Materials Corrosion and Protection Data Center, Institute for Advanced Materials and Technology, University of Science and Technology Beijing, Beijing 100083, China; cww1995210xy@sina.com (W.C.);; 2BRI Southeast Asia Network for Corrosion and Protection (MOE), Shunde Graduate School of University of Science and Technology Beijing, Foshan 528399, China; 3Shanghai Aerospace Equipments Manufacturer Co., Ltd., Shanghai 200245, China; 4State Key Laboratory of Metal Material for Marine Equipment and Application, Anshan 114002, China; 5Institute for the Development and Quality Macau, Macau 999078, China

**Keywords:** microbiologically influenced corrosion, extracellular electron transfer, stainless steels, *Shewanella algae*

## Abstract

The microbiologically influenced corrosion of 201 stainless steel by *Shewanella algae* was investigated via modulating the concentration of fumarate (electron acceptor) in the medium and constructing mutant strains induced by Δ*OmcA*. The ICP-MS and electrochemical tests showed that the presence of *S. algae* enhanced the degradation of the passive film; the lack of an electron acceptor further aggravated the effect and mainly affected the early stage of MIC. The electrochemical tests and atomic force microscopy characterization revealed that the ability of Δ*OmcA* to transfer electrons to the passive film was significantly reduced in the absence of the c-type cytochrome *OmcA* related to EET progress, leading to the lower corrosion rate of the steel.

## 1. Introduction

Microbiologically influenced corrosion (MIC) refers to the phenomenon that biofilm colonization, including bacteria, fungus and even archaea, at the metal/electrolyte interface influences the kinetic process of the anodic or cathodic reactions, subsequently altering the corrosion rate of the metal [1,2,3]. Extensive research has confirmed that the occurrence of large engineering accidents in various fields, including but not limited to the aerospace industry, oil pipeline industry and offshore drilling platforms, is directly or indirectly related to MIC [4,5]. In-depth exploration of the mechanisms behind MIC can effectively help in avoiding significant economic losses. Given that MIC solely occurs within minute areas of microbial colonization or beneath biofilms, there are still challenges in fully explaining its mechanisms. The traditional theory of cathodic depolarization has long been employed to explain MIC phenomena caused by sulfate-reducing bacteria (SRB) [6,7]. With the utilization of advanced characterization techniques, various mechanisms of MIC have been proposed, including the secretion of acidic metabolites, the formation of differential oxygen concentration cells and the extracellular electron transfer (EET) between the microorganisms and a metal surface [8,9,10].

So far, there are two recognized modes of the EET process: direct electron transfer (DET) and mediated electron transfer (MET). DET usually occurs via c-type cytochromes or conductive nanowires to make contact with the metal [11]. MET is achieved through the exchange of electrons between an external metal and microorganisms by soluble redox mediators or electron shuttles, such as flavonoids and quinones [12]. In recent years, EET has been recognized as a prevalent mechanism of MIC. It involves the entry of electrons generated from metal oxidation into the cytoplasm and completes the respiratory processes of microorganisms [13,14,15,16]. In addition, an increasing number of research studies suggested that microorganisms could utilize alloying compounds within passive film of stainless steel as alternative electron acceptors, which led to the damage of the passive film [17,18]. For example, Huang et al. demonstrated that a decrease in the concentration of NO^3−^, which served as the electron acceptor, accelerated the MIC of 304 stainless steel caused by *Pseudomonas aeruginosa*. Additionally, they used scanning electrochemical microscopy (SECM) to provide a microscopic electrochemical perspective on how the electron shuttle (PYO) was involved in the MIC of 304 stainless steel [19]. Hu et al. manipulated carbon source and electron acceptor concentrations in the environment and revealed that *Bacillus subtilis* primarily caused the breakdown of the passive film on the stainless steel surface through EET, leading to the accelerating of the corrosion [20].

*Shewanella* genera, as dissimilatory metal-reducing bacteria (DMRB), have been intensively investigated due to the unique respiratory versatility [21]. They reduce a wide range of organic compounds, polyvalent metal ions and solid metal oxides, including fumarate, nitrate, Fe (III), Mn (IV) and U (VI), as a part of their respiratory metabolism to obtain energy under anaerobic conditions [22]. The Mtr pathway including four multiheme c-type cytochromes (*CymA*, *MtrA*, *MtrC* and *OmcA*) and a porin-like outer membrane protein (*MtrB*) was reported to be associated with the ability to utilize such a wide array of electron acceptors [23,24]. *MtrC* and *OmcA* are both exposed on the outer membrane surface of the cell, allowing them to come into contact with extracellular soluble and insoluble electron acceptors [25]. The electrons from the interior of cells flow through the menaquinol pools, *CymA*, *MtrA* and *MtrB*, and finally through *MtrC* and *OmcA* to reach the extracellular electron acceptor [26]. A common approach in current research to investigate the role of EET in MIC is to knock out EET-related genes in bacteria and compare with the wild-type strain [27]. Li et al. conducted a study comparing the corrosion behavior of 304 stainless steel caused by a genetically manipulated strain (Δ*OmcA*) and the wild-type strain of *Shewanella oneidensis* MR-1 and confirmed that the absence of the *OmcA* significantly impeded the EET process [28]. Reardon et al. characterized the ability to reduce ferrihydrite of *Shewanella oneidensis* MR-1 mutants with in-frame deletions of *MtrC* and *OmcA* and demonstrated that outer-membrane cytochromes support a role for electron transfer from cell to the mineral [29].

*Shewanella algae* is a common bacterium in marine environments and has been reported to have the ability to reduce and deposit soluble metal ions such as ferrum (III), platinum (IV) and palladium (II) from dilute solutions [30,31]. In this work, we elucidated the role of EET in the MIC of 201 stainless steel (SS) caused by *S. algae* through modulating the concentration of electron acceptors in the culture medium and constructing mutant strains (Δ*OmcA*) lacking the gene encoding the cytochrome *OmcA*. Scanning electron microscopy (SEM) coupled with an energy dispersive spectroscopy detector (EDS) were used to observe the morphology and the composition of the 201 SS surface after 7 days of immersion. The concentrations of Fe^2+^ in various media were obtained by inductively coupled plasma mass spectrometry (ICP-MS). The corrosion behavior was characterized by potentiodynamic polarization curves and electrochemical impedance spectroscopy (EIS). The surface topography and the contact potential difference (*V_CPD_*) distribution on the coupons immersed in the medium inoculated with Δ*OmcA* were obtained by atomic force microscopy (AFM).

## 2. Materials and Methods

### 2.1. Materials

The 201 SS coupons (10 mm × 10 mm × 3 mm) were used for all tests and the chemical composition of 201 SS is shown in Table 1. For electrochemical tests, the coupons were sealed with epoxy resin and only one working face with 1 cm^2^ was exposed. All the coupons were sequentially abraded from 240 to 3000 grit and further polished with polishing compound, followed by an ultrasonic cleaning in anhydrous ethanol. Before the tests, coupons were sterilized under UV irradiation for 30 min.

### 2.2. Bacterial Mutant and Culture Condition

*S. algae* (MCCC 1A11468) was purchased from the Marine Culture Collection of China (MCCC), which was incubated in a *Shewanella* basal medium (SBM) with the following ingredients: 5.6 g/L sodium lactate, 3.2 g/L disodium fumarate, 4.78 g/L HEPES, 0.5 g/L casamino acid, 0.46 g/L NH_4_Cl, 0.225 g/L K_2_HPO_4_, 0.225 g/L KH_2_PO_4_, 0.117 g/L MgSO_4_·7H_2_O, 0.225 g/L (NH_4_)_2_SO_4_ and 10 mL/L trace element solution (15 mg NTA, 1 mg MnCl_2_·4H_2_O, 3 mg FeSO_4_·7H_2_O, 1.7 mg CoCl_2_·6H_2_O, 1 mg ZnCl_2_, 0.4 mg CuSO_4_·5H_2_O, 0.05 mg AlK(SO_4_)_2_·12H_2_O, 0.05 mg H_3_BO_3_, 0.9 mg Na_2_MoO_4_, 1.2 mg NiCl_2_, 0.2 mg NaWO_4_·2H_2_O and 1 mg Na_2_SeO_4_). The 50% fumarate group meant 1.6 g/L disodium fumarate in the medium. Before autoclaving at 121 °C for 20 min, the pH of the culture medium was adjusted to 7.05 ± 0.05. The medium was then deoxygenated by injecting pure nitrogen for 30 min. Subsequently, 1 mL seed culture of *S. algae* was inoculated in a 100 mL SBM and the flask was sealed with a rubber stopper in the anaerobic chamber. Three parallel coupons were set for each group of tests and the average value and standard deviation are shown in results. The optical density value at 600 nm (*OD*_600_) was measured by a UV spectrophotometer (Bio Mate3S, Thermo Fisher Scientific, Waltham, MA, USA) to reflect the growth of *S. algae*.

The mutant strains (Δ*OmcA*) lacking the gene encoding the cytochrome *OmcA* were constructed in the wild-type *S. algae* using homologous recombination techniques. Briefly, the first step was to design primers based on the flanking sequences of the target gene, and then perform the Polymerase Chain Reaction (PCR) amplification. The product was used as a template for the overlap extension PCR and the overlaps were spliced to obtain the sequence lacking the target gene. The sequence was inserted into the pRE112 vector using the restriction endonuclease SmaI-KpnI, completing the construction of the recombinant plasmid. The recombinant plasmid was then transformed into the *Escherichia coli* S17 λpir and positive colonies were selected using chloramphenicol-containing agar plates. The positive colonies were mixed with *S. algae* in a 1:1 ratio so that the target gene deletion was achieved through biparental conjugation. Finally, the mutant strains (Δ*OmcA*) were obtained through a two-step selection using chloramphenicol and 10% (*w*/*v*) sucrose agar plates [32].

### 2.3. Surface Analysis

After 7 days of immersion, the morphology of the biofilm on 201 SS coupons was characterized by SEM (Gemini 500, Carl Zeiss, Oberkochen, Germany) coupled with EDS (X-Max, Oxford, UK). The coupons were rinsed with phosphate buffer solution (PBS) to remove planktonic bacteria from the surface and then immersed in a 2.5 vol.% glutaraldehyde solution at 4 °C for 8 h to fix the cells. Subsequently, the cells were sequentially dehydrated by immersion in ethanol solutions of increasing concentrations (each for 10 min): 50%, 60%, 70%, 80%, 90%, and finally, 100%. The coupons were further sputter-coated with Au to improve the conductivity. The maximum diameters and depths of the corrosion pits on the coupons’ surfaces were measured by a white light interferometer (Contour X200, Bruker, Billerica, MA, USA).

### 2.4. Electrochemical Tests

The electrochemical tests were performed using an electrochemical workstation (CHI 660E, CH Instruments, Austin, TX, USA). A conventional three-electrode system was employed, containing a 201 SS coupon serving as the working electrode, a saturated calomel electrode (SCE) as the reference electrode and a graphite electrode as the counter electrode. All potentials in this paper were relative to SCE. The open circuit potential (OCP) test was first conducted for 10 min to ensure the stability of the system. The EIS test was performed with a sinusoidal wave perturbation of ±10 mV relative to OCP within a frequency range of 100 kHz to 10 mHz. EIS results were fitted by using the ZSimpWin software (Version 3.50). After 7 days of immersion, the potentiodynamic polarization test was conducted from −250 mV vs. OCP and scanning at a rate of 1 mV/s until the current density reached 0.1 mA/cm^2^.

### 2.5. ICP-MS Analysis

The concentration of Fe^2+^ in the culture medium was measured by ICP-MS (ICPOES730, Agilent, Santa Clara, CA, USA) after 7 days of immersion. All the media were pretreated with concentrated nitric acid (65 wt.%) at 80 °C for 30 min before the ICP-MS test.

### 2.6. AFM Characterization

AFM (Dimension ICON, Bruker) was employed to characterize the surface morphology of the coupons after 3 days of immersion, as well as *V_CPD_* between the bacteria or the coupon and the probe tip. A Cr/Co-coated silicon tip (HQ: NSC18/Co-Cr/Al BS, MikroMasch, Tallinn, Estonia) with a resonant frequency of 65.3 kHz and the force constant of 2.12 N/m was selected. The relationship between the *V_CPD_* and the work function (*WF*) can be calculated as follows when drive routing was set to sample [33]:(1)$$V_{CPD}=({WF}_{sample}-{WF}_{tip})/e$$
where *WF_sample_* and *WF_tip_* represents the work functions of the sample and the tip, and *e* is electron charge. All tests were conducted in air environment.

## 3. Results and Discussion

### 3.1. Bacterial Growth

Figure 1a shows the growth of *S. algae* over a 7-day incubation in different culture media. In the 100% fumarate medium, the *OD*_600_ value of *S. algae* gradually increased throughout the 7-day cultivation period. In the 50% fumarate medium, the *OD*_600_ value of *S. algae* was similar to that in the 100% fumarate medium on the first day and there was a gradual decline as the cultivation time progressed. According to the analysis of variance, *p* < 0.05 indicated a significant difference between the two groups. There was a significant difference in the number of bacteria after 7 days of immersion. This was due to the lower concentration of electron acceptors in the solution, which slightly restricted the EET activity of the planktonic bacteria and limited the energy acquisition and growth. Figure 1b depicts the variation of pH value for different media. The pH value of both media slightly decreased over the time, which related to little acidic microbial metabolites. The near-neutral environment eliminated the interference of acidic microbial metabolites in the corrosion of the steels. The pH value of the two media did not differ significantly over the 7-day immersion.

### 3.2. Surface Morphology

Figure 2 shows the SEM images of the 201 SS coupons immersed in the 100% fumarate medium inoculated with *S. algae*. After 1 day of immersion, the bacteria dispersed on the 201 SS surface and aggregated on local areas of the surface (Figure 2a). As the immersion time reached 3 days, the density of bacteria on the surface increased (Figure 2b). After 7 days of immersion, the number of sessile bacteria further increased and localized dense biofilms formed (Figure 2c). Some corrosion pits were observed around the sessile bacteria on the coupon’s surface (Figure 2d and Figure 3a). Within the area spanning the corrosion pit (white dashed line), an EDS line scan was performed on Fe, Cr and O elements and the results are shown in Figure 3b–d. The O element content in the corrosion pit area was only 20% of that in the other regions, while the corresponding Fe and Cr element contents were increased by approximately 15%. The line scan results were caused by the rupture of the passive film on the 201 SS surface, which led to the exposure of the underlying substrate.

### 3.3. ICP-MS Results

Figure 4 presents the concentration of Fe^2+^ in the sterile and inoculated media after 7 days of immersion. After immersing in the sterile medium for 7 days, the concentration of Fe^2+^ was approximately 14 μg/L, which indicated that the 201 SS only slightly corroded under abiotic conditions. Correspondingly, the concentration of Fe^2+^ in the 100% fumarate and 50% fumarate media under biotic conditions reached 58 μg/L and 78 μg/L, respectively. The significant higher concentration of the Fe ion leaching confirmed that the presence of *S. algae* could accelerate the corrosion of 201 SS in an anaerobic environment and the MIC was further enhanced with the lack of electron acceptor.

### 3.4. Electrochemical Analysis

Figure 5 shows the potentiodynamic polarization curves of the coupons in the sterile medium and the inoculated media with various fumarate concentrations after 7 days of immersion. After immersing in the sterile medium for 7 days, the passivation current density (*i_pass_*) of 201 stainless steel was 0.13 μA and the pitting potential (*E_pit_*) was 0.76 V, indicating good corrosion resistance of the steel in the sterile medium. In comparison to the sterile medium, the *i_pass_* of the coupon in the 100% fumarate medium inoculated with *S. algae* increased to 3.3 μA and *E_pit_* decreased to 0.37 V. When the electron acceptor in the culture medium was halved, the *E_pit_* of the 201 SS coupon further decreased to 0.15 V, while *i_pass_* increased to 7.2 μA. The results from the potentiodynamic polarization curves indicate that *S. algae* cells accelerated the degradation of the passive film. The lack of an electron acceptor made the biofilm more corrosive to the passive film. These results were consistent with the ICP results.

To comprehensively understand the effect of electron acceptor concentration on the MIC of stainless steel caused by *S. algae*, a non-destructive EIS test was used to observe the corrosion behavior of 201 SS in the 100% fumarate- and 50% fumarate-inoculated media at different immersion times (Figure 6). The diameter of the capacitive loops of the steel in the 50% fumarate medium was consistently smaller than that in the 100% fumarate medium at the same immersion time. The diameter of the capacitive loops in the low-frequency region of the Nyquist plots showed a significant decrease after 1 day of immersion in both media, followed by a slight increase with prolonged immersion time. Correspondingly, the Bode plots showed that the impedance modulus in the low-frequency region (*|Z|_0.01 Hz_*) exhibited consistent variation patterns as capacitive loops. The larger diameter of a capacitive loop and higher *|Z|_0.01 Hz_* value are commonly used as indicators of better corrosion resistance [34]. Two time constants were observed in the phase angle plots of the coupons immersed in both media, including one for the passive film with biofilm attachment and the other for the charge transfer process of the electrical double layer formed at the interface of the metal substrate and the medium.

The equivalent electrical circuit in Figure 7a was used to fit the EIS results of coupons immersed in both media for 3 h, 1 day and 3 days. With the thickening of the biofilm, resulting in the increase in *|Z|_0.01 Hz_* and the diameter of the capacitive loops, the equivalent electrical circuit in Figure 7b was used to fit the results after 5 and 7 days of immersion. The constant phase element *Q* is usually used instead of a pure capacitance to fit non-ideal films on the surface. *R_s_* represents the solution resistance. *Q_f_* and *R_f_* are related to the capacitance and resistance of the surface film consisting of the biofilm and the passive film, respectively. *Q_dl_* is the capacitance of the electrical double layer and *R_ct_* reflects the resistance of the charge transfer resistance.

Table 2 summarizes the fitting results for coupons in the inoculated media. After 1 day of immersion, the coupons immersed in the 100% fumarate medium exhibited a decrease in *R_ct_* from (4.95 ± 0.89) × 10^5^ Ω cm^2^ to (2.81 ± 0.63) × 10^4^ Ω cm^2^, while that immersed in the 50% fumarate medium showed a decrease from (3.38 ± 0.66) × 10^5^ Ω cm^2^ to (2.70 ± 0.48) × 10^4^ Ω cm^2^. As the immersion time increased, the *R_ct_* value gradually increased in both media, which is associated with the expansion and thickening of the biofilm. The results indicated that the attachment and growth of bacteria during the initial immersion period induced the degradation of the passive film and accelerated the corrosion of 201 SS. The decrease in the concentration of electron acceptors in the medium led to the preference of *S. algae* to utilize the passive film as the alternative electron acceptor, thereby accelerating the degradation of the passive film. With the extension of immersion time, the thickening of biofilm prevented 201 SS from direct contact with the solution and reduced part of the charge transfer, which increased *R_f_* and *R_ct_* values and slightly inhibited the corrosion of the steel.

Figure 8 presents the morphology of the maximum corrosion pits on the 201 SS surface after immersion for 7 days in 100% fumarate and 50% fumarate media. In the 100% fumarate medium, the width of the maximum corrosion pit was 17.87 µm, whereas the depth was 1.32 µm. Notably, the steel surface immersed in the 50% fumarate medium showed a much wider (33.40 µm) but shallower (0.72 µm) pit. These results suggested that the absence of fumarate primarily promoted the interaction between *S. algae* and the oxides within the passive film, leading to the formation of wider corrosion pits. However, the absence of fumarate does not exacerbate the MIC of the metallic substrate.

### 3.5. Corrosion Behavior of ΔOmcA

Figure 9 presents the SEM images of the 201 SS coupons immersed in the 100% fumarate medium inoculated with Δ*OmcA*. It could be observed that Δ*OmcA* attached in large quantities to the surface and formed localized biofilms on the first day of immersion (Figure 9a). By the third day, the biofilm coverage area increased and thickened (Figure 9b). After 7 days of immersion, the biofilm almost covers the entire surface of the 201 SS (Figure 9c). Compared to the wild-type *S. algae*, Δ*OmcA* had a shorter length and a higher number of sessile cells on the steel and few corrosion pits could be observed around the cells (Figure 9d). The SEM results showed that the growth of Δ*OmcA* was not affected by knocking out related genes and the number of sessile cells was more than *S. algae*. 

Figure 10 presents the potentiodynamic polarization curve of the 201 SS coupon in the medium inoculated with Δ*OmcA* after 7 days of immersion. Compared to the coupon immersed in the sterile medium, the coupon immersed in the Δ*OmcA*-inoculated medium with 100% fumarate showed nearly identical *E_pit_* (0.74 V), which was significantly higher than that immersed in the *S. algae*-inoculated medium (0.37 V). The coupon in the Δ*OmcA*-inoculated medium had an *i_pass_* of 1.12 μA, which falls between the *i_pass_* values of coupons immersed in the sterile medium and the *S. algae*-inoculated medium. Δ*OmcA* cells slightly accelerated the MIC of the steel, but it was much weaker than that mediated by *S. algae*.

Figure 11 shows the Nyquist and Bode plots of 201 SS in the 100% fumarate medium inoculated by Δ*OmcA* at different immersion times. In the Nyquist plots, the diameter of the capacitive loops in the low-frequency region continuously decreased during the initial immersion period, which increased by the 5th and 7th days of immersion. The same equivalent electrical circuit in Figure 7a was used to fit the EIS results after 3 h of immersion and that in Figure 7b was used to fit the remaining results. The fitting results are summarized in Table 3. After 1 day of immersion, the *R_ct_* of the coupons in the medium inoculated with Δ*OmcA* decreased from (9.84 ± 0.17) × 10^6^ Ω cm^2^ to (1.78 ± 0.08) × 10^6^ Ω cm^2^. It reached the lowest value of (1.75 ± 0.12) × 10^5^ Ω cm^2^ after 3 days of immersion. This indicated that Δ*OmcA* had a certain degree of corrosion-accelerating effect on the steel during the first 3 days of immersion, but this effect was significantly weaker than that of *S. algae* (Figure 6). When the lowest corrosion resistance of 201 SS induced by Δ*OmcA* appeared, it took a longer time and the corrosion rate was slower. Based on the electrochemical test results, it was evident that *S. algae* obviously exacerbated the degradation of the passive film. However, when the protein *OmcA* which is associated with the electron transfer pathway was absent, the MIC of the steel was noticeably weakened.

In our recent study [35], we found that the *V_CPD_* drop of the passive film occurred at the locations of the adherent *S. algae* using AFM and suggested that the *V_CPD_* drop should be resulted from the electrons injected into the passive film from cells. Herein, AFM was also used to further investigate the potential around Δ*OmcA* on the 201 SS surface after 3 days of immersion as shown in Figure 12. The morphology and sizes of the cells in Figure 12a were consistent with the SEM observation (Figure 9). In the *V_CPD_* distribution map, no *V_CPD_* drop region was observed in the vicinity of the bacteria (Figure 12b). The area outlined by the white dashed line was selected for detailed analysis (Figure 12c,d). The *V_CPD_* of the passive film around Δ*OmcA* was almost the same as that without bacterial attachment, indicating that the ability of bacteria to transfer electrons to the passive film was significantly reduced in the absence of the c-type cytochrome *OmcA*. This also caused a slow degradation rate of the passive film, and the MIC of stainless steel was inhibited. 

## 4. Conclusions

This work investigated the role of EET in the MIC of 201 stainless steel caused by *S. algae* via modulating the concentration of electron acceptor in the medium and constructing mutant strains by Δ*OmcA*. The major conclusions were obtained as follows:According to the results of the potentiodynamic polarization curves and the ICP-MS, 201 stainless steels corroded slowly in the sterile medium. The presence of *S. algae* enhanced the dissolution of the Fe oxides in the passive film and decreased the pitting potential of the steel, leading to the degradation of the passive film.The results of EIS and the morphologies of the maximum corrosion pits revealed that corrosion acceleration of 201 stainless steel was most obvious in the 50% fumarate medium in the presence of *S. algae*. The lack of an electron acceptor further aggravated the degradation of the passive film and mainly affected the early stage of MIC.In the absence of the c-type cytochrome *OmcA*, MIC of the 201 stainless steels induced by Δ*OmcA* was significantly weakened as manifested by the potentiodynamic polarization curves, EIS and AFM characterization. It further confirmed the important role of *OmcA* related to EET in the MIC of 201 stainless steel.

## Figures and Tables

**Figure 1 materials-16-05209-f001:**
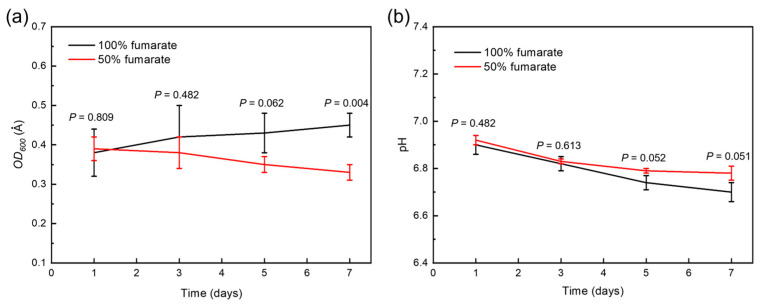
(**a**) Growth curves of *S. algae* and (**b**) pH variation in various fumarate concentration medium.

**Figure 2 materials-16-05209-f002:**
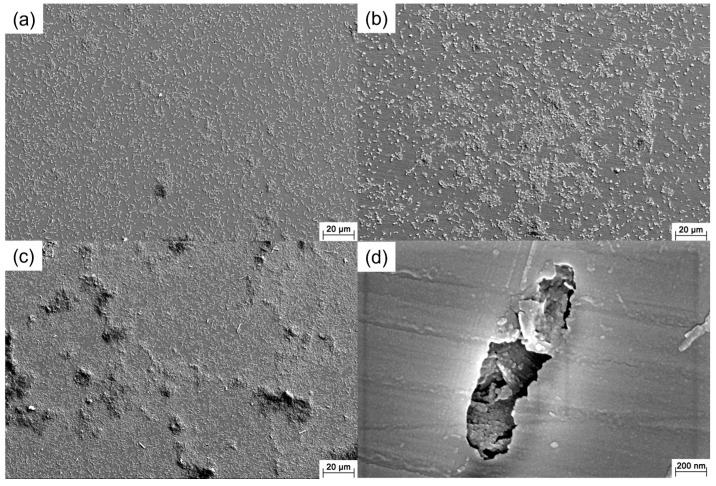
SEM images of the 201 SS immersed in the 100% fumarate medium inoculated with *S. algae* for (**a**) 1 day, (**b**) 3 days and (**c**,**d**) 7 days.

**Figure 3 materials-16-05209-f003:**
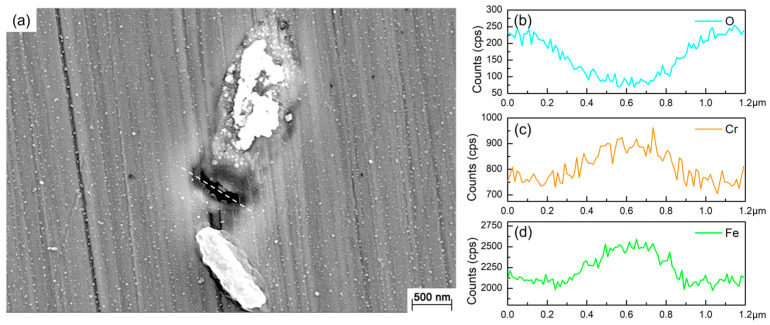
(**a**) SEM images of the corrosion pit around the sessile cell after 7 days of immersion in the 100% fumarate medium and the EDS line scan results of (**b**) O, (**c**) Cr and (**d**) Fe elements.

**Figure 4 materials-16-05209-f004:**
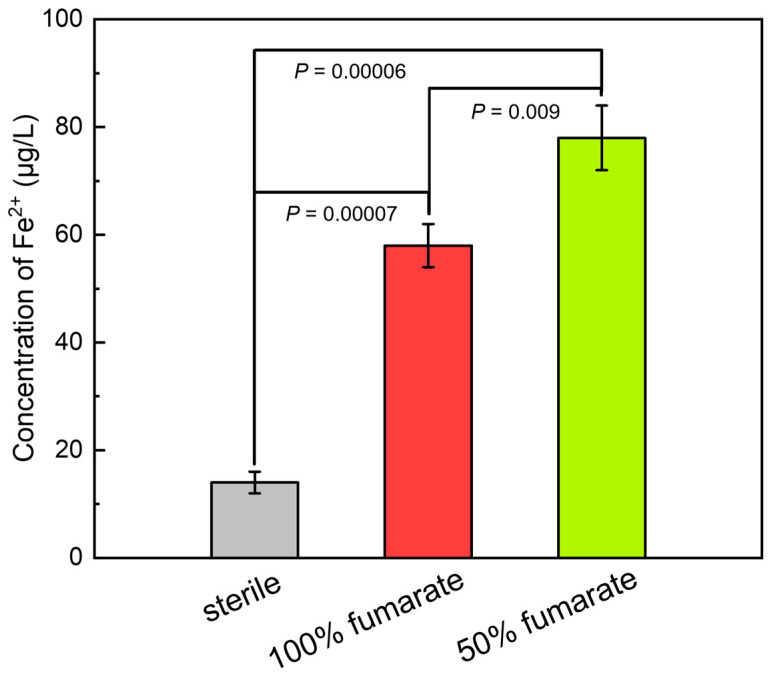
The concentration of Fe^2+^ after 7 days of immersion in various media. *p* < 0.05 indicated a significant difference between the two groups.

**Figure 5 materials-16-05209-f005:**
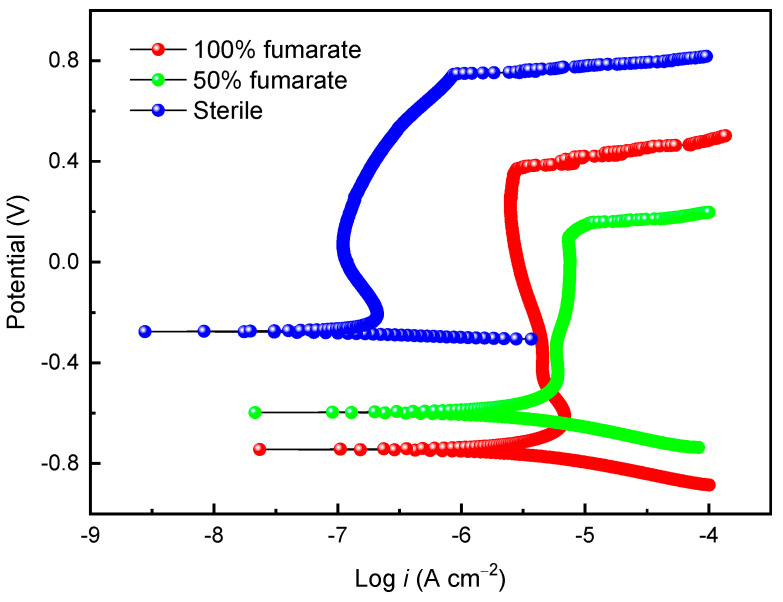
The potentiodynamic polarization curves of the 201 SS coupon immersed in the sterile medium and inoculated media containing *S. algae* with various fumarate concentrations.

**Figure 6 materials-16-05209-f006:**
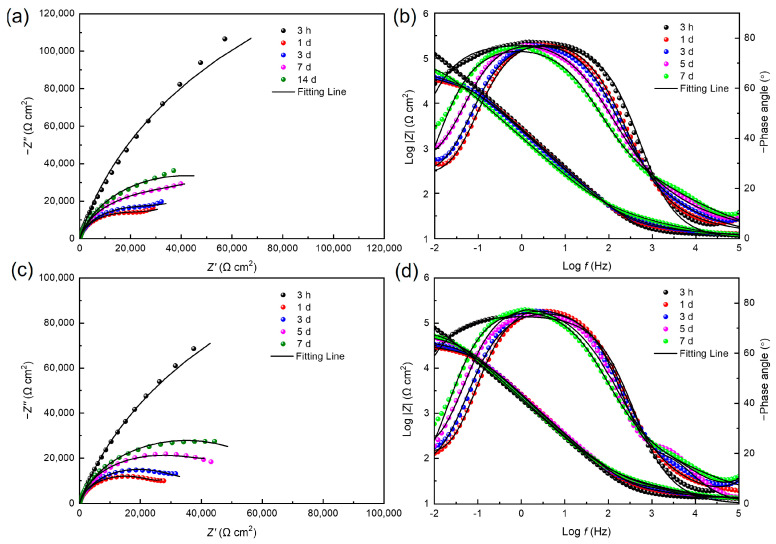
Nyquist and Bode plots for 201 SS coupons immersed in the (**a**,**b**) 100% fumarate medium and (**c**,**d**) 50% fumarate medium.

**Figure 7 materials-16-05209-f007:**
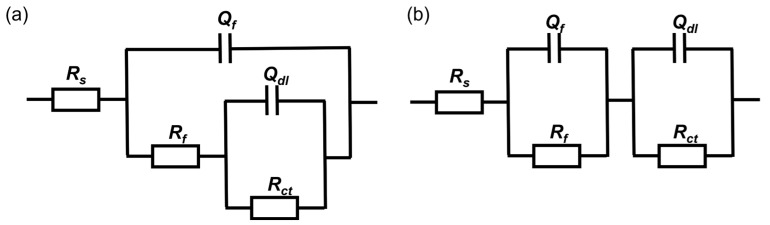
The equivalent electrical circuit used to fit the EIS spectra of coupons immersed in the inoculated media for (**a**) 3 h, 1 day, 3 days, (**b**) 5 days and 7 days.

**Figure 8 materials-16-05209-f008:**
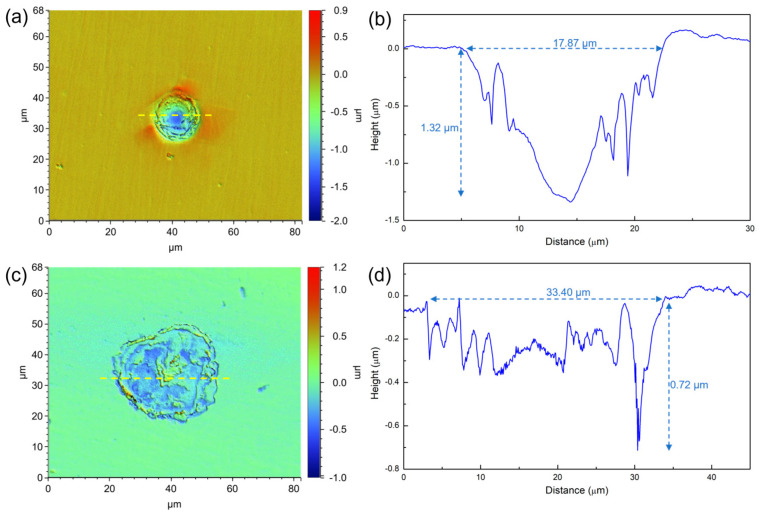
The morphology of the maximum corrosion pit on 201 SS coupons after 7 days of immersion (**a**,**b**) in the 100% fumarate medium and (**c**,**d**) the 50% fumarate medium. The yellow dashed lines mark the location of the line scan results of depth. The blue dashed lines mark the width and the depth of corrosion pits.

**Figure 9 materials-16-05209-f009:**
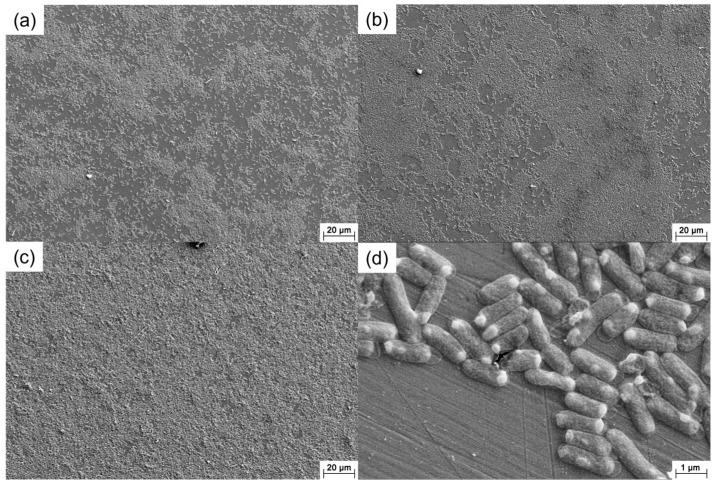
SEM images of the 201 SS immersed in the 100% fumarate medium inoculated with Δ*OmcA* for (**a**) 1 day, (**b**) 3 days and (**c**,**d**) 7 days.

**Figure 10 materials-16-05209-f010:**
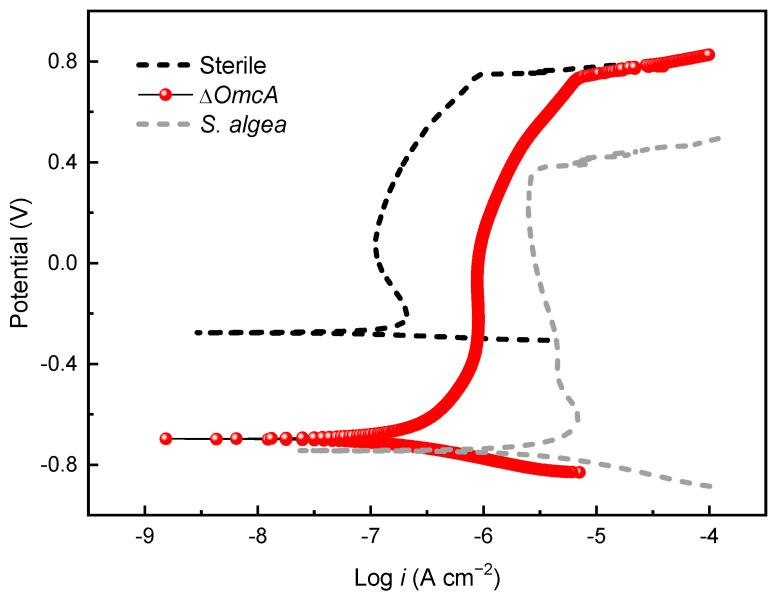
The potentiodynamic polarization curve of the 201 SS coupon immersed in the 100% fumarate medium inoculated with Δ*OmcA* after 7 days.

**Figure 11 materials-16-05209-f011:**
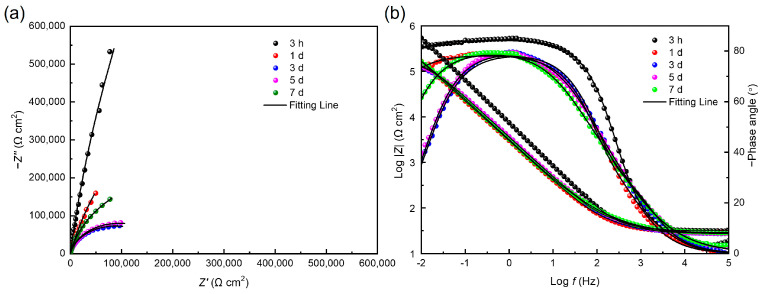
(**a**) Nyquist and (**b**) Bode plots for 201 SS coupons immersed in Δ*OmcA*-inoculated medium.

**Figure 12 materials-16-05209-f012:**
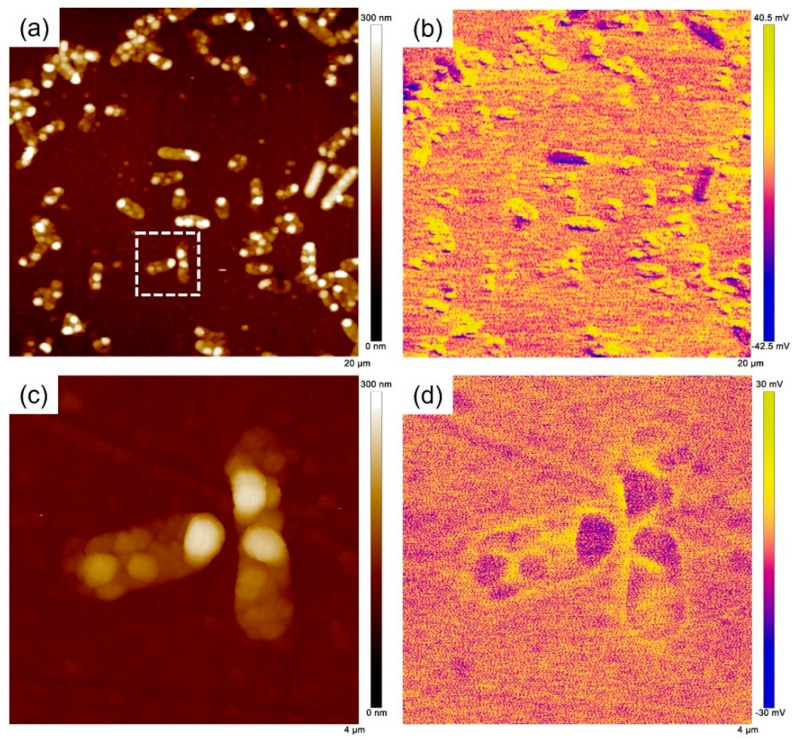
(**a**) AFM topography and (**b**) *V_CPD_* potential distribution on 201 SS surface after 3 days of immersion. (**c**,**d**) The corresponding results of the enlarged area are outlined by the white dashed line.

**Table 1 materials-16-05209-t001:** The chemical composition of 201 SS coupons (wt.%).

Element	C	Cr	Mn	Si	Ni	S	P	Fe
201 SS	0.12	17.20	6.20	0.80	4.80	0.01	0.03	balance

**Table 2 materials-16-05209-t002:** EIS parameters of 201 SS coupons in the 100% fumarate medium and the 50% fumarate medium inoculated with *S. algae*.

Time	*R_s_* (Ω cm^2^)	*Q_f_* × 10^−5^ (Ω^−1^ cm^−2^ s^n^)	*R_f_* (Ω cm^2^)	*Q_dl_* × 10^−5^ (Ω^−1^ cm^−2^ s^n^)	*R_ct_* × 10^4^ (Ω cm^2^)
100% fumarate medium					
3 h	14.52 ± 2.32	2.26 ± 0.76	40.36 ± 4.2	5.19 ± 0.32	49.5 ± 8.9
1 d	12.05 ± 1.82	2.49 ± 0.30	11.7 ± 3.8	5.01 ± 0.42	2.81 ± 0.63
3 d	12.70 ± 1.52	2.42 ± 0.29	15.1 ± 3.3	5.75 ± 0.52	3.47 ± 0.46
5 d	10.12 ± 1.24	2.41 ± 0.47	22.1 ± 5.2	6.48 ± 0.64	5.76 ± 0.97
7 d	9.68 ± 1.01	2.61 ± 0.36	49.3 ± 7.9	1.19 ± 0.08	8.40 ± 1.12
50% fumarate medium					
3 h	10.33 ± 1.32	2.10 ± 0.42	4.89 ± 0.10	8.72 ± 0.76	33.8 ± 6.6
1 d	13.92 ± 1.44	2.25 ± 0.37	11.7 ± 2.6	5.05 ± 0.62	2.70 ± 0.48
3 d	13.73 ± 1.11	2.21 ± 0.29	15.5 ± 2.9	5.45 ± 0.77	3.41 ± 0.46
5 d	14.04 ± 1.03	2.28 ± 0.44	22.0 ± 6.2	5.66 ± 0.40	5.12 ± 0.95
7 d	10.39 ± 0.92	1.80 ± 0.09	46.5 ± 8.3	9.06 ± 0.82	6.59 ± 1.02

**Table 3 materials-16-05209-t003:** EIS parameters of 201 SS coupons in the 100% fumarate medium and the 50% fumarate medium inoculated with Δ*OmcA*.

Time	*R_s_* (Ω cm^2^)	*Q_f_* × 10^−5^ (Ω^−1^ cm^−2^ s^n^)	*R_f_* (Ω cm^2^)	*Q_dl_* × 10^−5^ (Ω^−1^ cm^−2^ s^n^)	*R_ct_* × 10^5^ (Ω cm^2^)
Δ*OmcA*					
3 h	13.16 ± 1.77	0.88 ± 0.04	16.6 ± 3.6	1.54 ± 0.63	98.4 ± 1.7
1 d	12.95 ± 2.01	3.56 ± 0.64	11.7 ± 3.8	3.13 ± 0.45	17.8 ± 0.8
3 d	12.79 ± 1.14	2.37 ± 0.12	15.1 ± 3.3	3.03 ± 0.62	1.75 ± 0.12
5 d	12.83 ± 1.62	2.48 ± 0.95	22.1 ± 5.2	2.97 ± 0.27	1.98 ± 0.43
7 d	12.77 ± 1.36	53.4 ± 2.2	49.3 ± 7.9	6.20 ± 0.72	5.10 ± 1.67

## Data Availability

The data presented in this study are available on request from the corresponding author.

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
