# Peer review of "Extracellular Electron Transfer in Microbiologically Influenced Corrosion of 201 Stainless Steel by Shewanella algae"

_materials, 2023, doi:10.3390/ma16155209_

Round 1

Reviewer 1 Report

Through the papers published so far, the authors show the continuity of research in the field of Microbiologically influenced corrosion.

In order to improve the quality of the manuscript:

Please provide the statistical analysis of Figure 1. (a) Growth curves of S. algae and (b) pH variation in various fumarate concentration medium and please include the statistical analysis in the Results and discussion.

In the chapter 3.3. ICP-MS results at Figure 4 some more information on the statistics and standard deviations or errors should be given.

Figures are of poor quality and need to be improved.

Author Response

Please find the document in the attachment.

Reviewer 2 Report

There is no comment.

Author Response

(The authors gave the same response as above.)

Reviewer 3 Report

It is a very interesting and novel work. It constitutes a contribution to the knowledge of the mechanism of corrosion induced by micro-organisms. In the introduction it is recommended that the definition of MIC be broader and not limit it only to bacteria. It is suggested to change bacteria by micro-organisms. The resolution of some figures should be improve.

Author Response

(The authors gave the same response as above.)
